# Benefits of Urban Forest Healing Program on Depression and Anxiety Symptoms in Depressive Patients

**DOI:** 10.3390/healthcare11202766

**Published:** 2023-10-19

**Authors:** Poung-Sik Yeon, Si-Nae Kang, Nee-Eun Lee, In-Ok Kim, Gyeong-Min Min, Ga-Yeon Kim, Jin-Gun Kim, Won-Sop Shin

**Affiliations:** 1Department of Forest Sciences, Chungbuk National University, Cheongju 28644, Republic of Korea; well@chungbuk.ac.kr; 2Graduated Department of Forest Therapy, Chungbuk National University, Cheongju 28644, Republic of Korea; sinae375@gmail.com (S.-N.K.); share1227@gmail.com (N.-E.L.); inoya88@naver.com (I.-O.K.); akwjs5019@chungbuk.ac.kr (G.-M.M.); rkdus6520@naver.com (G.-Y.K.); 3Korea Forest Therapy Forum Incorporated Association, Cheongju 28644, Republic of Korea

**Keywords:** urban forest healing program, depression, anxiety

## Abstract

Depression is considered a widespread mental health problem worldwide. Moreover, anxiety symptoms are very closely related to depression in patients, and it is known that the coexistence rate of depression and anxiety diagnosed simultaneously is high. Treatment and preventive management of depression and anxiety are essential for public health. Forest healing is attracting attention as a form of low-cost preventive medicine that is safe and has no side effects. However, although the physiological and psychological effects have been scientifically proven, it is insufficient to reveal a direct relationship between forest healing and depression. This study investigated the benefits of an urban forest healing program on depression and anxiety symptoms in depressive disorders. We employed a randomized controlled trial design. Forty-seven depressive patients were randomly divided into an urban forest healing program group and a control group. Measures included the Montgomery-Asberg depression rating scale (MADRS), the Hamilton Anxiety Rating Scale (HARS), and the State-Trait Anxiety Inventory (STAI) questionnaires. Our results revealed that the combination of general treatment and forest healing programs for patients with depression is more effective in improving depression and anxiety than routine treatment alone. We expect our work to serve as a starting point for more sophisticated research discussing the availability of non-pharmacological treatments in forest healing.

## 1. Introduction

Depression is considered a widespread mental health problem around the world. According to the World Health Organization [1], it is estimated that 3.8% of the world’s population suffers from depression, including 5.0% of adults and 5.7% of adults older than 60 years. In other words, depression affects about 280 million people. Depression is diagnosed based on anhedonia, combined with symptoms that can appear as key symptoms of depression, and emotional and neurocognitive symptoms, including suicidal thoughts, insomnia, and agitation [2].

Anxiety is often described as anxiety, fear, or doubt. Anxiety includes anxiety, fatigue, muscle tension, insomnia, and difficulty concentrating [3]. Moreover, depression and anxiety symptoms are very closely related, and it is known that the co-existence rate diagnosed simultaneously is high. About 85% of patients with depression experience significant anxiety symptoms, while 90% of patients with anxiety disorders are also reported to suffer from depression [4]. Since the coexistence of these two diseases increases the risk of suicide, the treatment and preventive management of depression and anxiety are crucial for public health.

The treatment of patients with depression consists of medication and non-pharmacological treatments. Medication for patients with depression includes selective serotonin reuptake inhibitors (SSRI), serotonin–norepinephrine reuptake inhibitors (SNRI), tricyclic agents, and benzodiazepine [5]. Among all the treatment options, medication is maintained as the first line [6]. However, the long-term side effects of antidepressants can negatively affect daily life due to gastrointestinal symptoms, neurological symptoms, and sexual dysfunction, limiting the excessive use of medication [7]. On the other hand, non-pharmacological treatments consist of cognitive behavioral therapy, psychotherapy, mindfulness-based therapy [8], exercise training, and music therapy [9,10]. For example, Verrusio et al. [9] investigated the impact of exercise training and music therapy on moderate depression patients. This study showed that subjects who participated in exercise training combined with listening to music improved their depression and anxiety symptoms. On the other hand, subjects who participated in pharmacotherapy improved only anxiety.

Forest healing is in the spotlight as one of these non-pharmacological interventions. Forest healing is an activity that improves mental and physical health by utilizing various elements of the forest, such as the landscape, sound, and phytoncide [11]. It has been scientifically proven that forest healing provides beneficial benefits for human psychological and physiological health [12,13,14]. Forest healing is recommended as a form of low-cost preventive medicine that is safe and has no side effects [15]. When exposed to forest environments, people unconsciously feel free and comfortable, and their minds are boosted and energized. The forest healing program is a program that combines various activities such as exercise, walking, breathing, and play activities using forest environmental factors to maximize the healing effect of the forest [16]. Many previous studies have found the positive effects of forest healing programs on mental health, such as reducing psychological stress or mental fatigue [17,18]. Among them, studies on the clinical effects of forest healing to improve depression and anxiety have been reported. For example, Chun et al. [19] showed that a three-night, four-day forest healing program conducted with 92 chronic alcohol abusers improved subjects’ depressive symptoms and anxiety. In addition, Choi et al. [20] documented that eight sessions of a forest healing program for cancer patients reduced depression in cancer patients more than daily activities. Kim et al. [21] also reported that a 3-day forest healing program reduced depression and anxiety compared to daily activities. Lim et al. [22] showed that 11 sessions of a forest healing program conducted with 64 people aged 65 or older significantly improved depression in the elderly. Han et al. [23] reported that a one-night, two-day forest healing program conducted with 61 workers reduced their depressive symptoms. In addition, previous meta-analyses evaluating the impact of forest healing on depression and anxiety confirmed that forest healing is a very effective intervention in improving depression and anxiety [24,25,26].

However, although many studies have reported that forest healing reduces subjects’ depression and anxiety, previous studies are insufficient to reveal a direct link between forest healing and depression. The existing studies have evaluated the improvement in depression and anxiety through forest healing in diseased groups or healthy people such as in cases of cancer or alcohol abuse, not depression patients. In addition, numerous previous studies have been conducted in deep forests with less accessibility. It is necessary to verify the program’s effectiveness using forests near the city regarding continuous health care for depressed patients. Therefore, this study was conducted with the hypothesis that forest healing programs using urban forests could alleviate depression and anxiety symptoms in depressed patients.

In this study, the following research hypothesis was established.
The urban forest healing program group will have fewer depression symptoms (on the MADRS) than the control group.The urban forest healing program group will have fewer anxiety symptoms (on the HARS and STAI-T) than the control group.

## 2. Materials and Methods

### 2.1. Study Area

We conducted this study in Seoul Forest. Seoul Forest is located in Seondong-gu, Seoul Metropolitan City, Republic of Korea (Figure 1). With a total area of 1,156,498 square meters, it is Seoul’s third largest urban park after Mapo-gu World Cup Park (3,305,785 square meters) and Songpa-gu Olympic Park (1,652,892 square meters). Seoul Forest consists of four distinctive spaces: a culture and art park, an experiential learning center, an ecological forest, and a wetland ecological center, and it is in contact with the Han River, providing various cultural leisure spaces. The forest tree species are mainly composed of conifers (such as *Pinus strobus*, Ginkgo biloba, Pinus densiflora, and Metasequoia glyptostroboides) and broadleaf trees such as *Zelkova serrata*, *Prunus serrulate*, Cercidiphyllum japonicum, and Malus Pumila Crataegus pinnatifida. The weather during the periods of the experiment was sunny, and the average temperature was 21.9 °C.

### 2.2. Subjects

A sample size calculation using G*Power 3.1 (University of Düsseldorf, Düsseldorf, Germany) resulted in 52 subjects. We added three more subjects to ensure that potential dropouts would not reduce the sample size. The total sample size consisted of 55 subjects in this study. Subjects were recruited from four specialized mental health medical clinics in Seoul, Korea. Experienced psychiatrists within the mental health medical clinics informed visiting depressive patients about this study. Overall, sixty subjects participated from March to April 2022. Subjects were eligible for inclusion if they were (a) diagnosed with mild depressive disorder by psychiatrists following DSM-5 and identified through a structured clinical interview (SCID) for DSM disorders in the Korean version, (b) aged 20 to 60 years, and (c) patients with the potential for outdoor activities of more than 2 h. In contrast, exclusion criteria were (a) people who had participated in a program similar to this forest healing program within the last three months and (b) those who are allergic to pollen or suffer from physical diseases other than depression.

Fifty-five subjects were recruited for the study, but five dropped out due to personal problems. So, fifty initial subjects were randomly assigned to either the urban forest healing program (25 subjects) or the outpatient control group (25 subjects). During the experiments, three subjects were suspended from the urban forest healing program group for health and personal reasons, and no subjects were eliminated from the control group. Therefore, the urban forest healing program group that completed this study consisted of 22 subjects, including 7 men and 15 women, with an average age of 37.8 ± 10.3 years old. The outpatient control group consisted of 25 subjects, 1 man and 24 women, and the average age was 38.9 ± 10.5 years. The study’s purpose and procedures were explained to all the subjects before the experiment. Also, the subjects did not know whether their group was an experimental group or a control group, and they were informed in advance that the program period was divided into two stages and asked for consent.

The study was approved by the Institutional Review Board of Chungbuk National University (CBNU-202203-HR-0041). We obtained written informed consent from all the subjects and provided USD 200.00 compensation for the subjects in this study.

### 2.3. Design

This study was designed as a randomized controlled clinical trial in which an individual experimental group (urban forest healing program) was compared to a control group (treatment as usual). Randomization was executed per subject by a computer random number generator. We created a list of random numbers from a minimum of 1 to a maximum of 50. The experimental group was assigned odd numbers, and the control group was assigned even numbers. A total of 50 subjects were randomly allocated to the experimental and control groups, with an allocation ratio of 1:1. The experiment was conducted between 10 a.m. and 12 p.m. from 12 May to 21 June 2022.

### 2.4. Composition of Urban Forest Healing Program

The urban forest healing program for depressed patients was developed by four licensed forest healing instructors and two psychiatrists. The urban forest healing program consisted of six sessions, and the time required for each session was 90 min. The intervention aims to reduce depressive and anxiety symptoms in adults with a depressive disorder. The modules of the core forest healing program are divided into three stages. The first step (first and second sessions) aims to explore and clarify feelings as a ‘recognition’ stage. The second step (third and fourth sessions) aims to stop negative thoughts through various activities in the urban forest as an ‘action’ stage. The third step (fifth and sixth sessions) aims to induce a break in the ring of ruminative thinking in the ‘change’ stages as the last step. We aimed to ultimately improve the lifestyle of depressed patients by repeatedly learning the activities of stretching exercises, walking and the five senses (sight, sound, smell, taste, and touch), an emotional card play game, and breathing and meditation, step-by-step, and practicing them in daily life.

### 2.5. Study Procedure

The experimental intervention was a structured forest healing program in an urban forest (Figure 2). All the urban forest healing programs were provided by licensed forest healing therapists. This experimental intervention involved three forest healing therapists, three research assistants, and one counselor. The experimental intervention was conducted once a week for 90 min for each group of 8–9 subjects and was conducted with the three groups for six weeks. The operation method of the urban forest healing program was to assign one forest healing therapist and one research assistant per group. The forest healing therapist helped the subjects effectively carry out forest healing activities according to the healing module stage. The research assistant placed the equipment necessary for the program at the place of the activity so that the forest healing program could be operated smoothly, and distributed and collected questionnaires during the evaluation. In addition, one counselor participated in the forest healing program to periodically assess the condition of the depressed patients.

The subjects in the control group did not receive any forest healing activities during the experiment and received treatment as usual. Treatment included medication and counseling as usual. We provided the same urban forest healing program to the control group after the study’s end, so that the control group’s subjects did not know whether they were in the experimental group or the control group.

All the subjects were continuously managed by a general physician. The subjects were maintained according to each patient’s previous treatment schedule for six weeks.

Data collection was conducted through pre-evaluation and post-evaluation. The pre-evaluation was conducted between 12 May 2023, and 13 May 2023, with MADRS and HARS examined by psychiatrists, and STAI-T, which required subjects to complete questionnaires. The psychiatrist did not know whether the subject was an experimental or a control group to remove the skewness of the intervention result measurement.

After the six-week intervention, the experimental group conducted a post-evaluation from 18 June to 21 June 2023, the same way as the pre-evaluation. In the case of the control group, the post-evaluation was conducted simultaneously with the time of the experimental group’s post-evaluation.

### 2.6. Measurements

The psychological evaluations used the Montgomery-Asberg depression rating scale (MADRS), the Hamilton Anxiety Rating Scale (HARS), and the State-Trait Anxiety Inventory (STAI) questionnaires. The MADRS was developed by Montgomery and Åsberg [27] and is a clinician-rated measure of depression severity. It consists of the following 10 items: (1) apparent sadness; (2) reported sadness; (3) inner tension; (4) reduced sleep; (5) reduced appetite; (6) concentration difficulties; (7) lassitude; (8) inability to feel; (9) pessimistic thoughts; and (10) suicidal thoughts. These items are clinician-rated on a seven-point Likert scale and are summed to produce a total scale score ranging from 0 to 60, with higher scores reflecting greater depression severity. Depression severity can be categorized into normal (0–6), mild depression (7–19), moderate depression (20–34), and severe depression (34–60). This study used the highly reliable Korean version of MADRS [28]. The K–MADRS of this study achieved a high reliability (Cronbach’s α = 0.96).

The HARS was developed by Hamilton [29] and is applied by the clinician to determine anxiety level and symptom distribution. It contains 14 questions, including sub-dimensions questioning both psychic and somatic symptoms. It is a five-point Likert-type scale (range 0–4). The total score is calculated by the sum of the scores obtained from each item. The severity of the anxiety can be categorized into normal (0–7), mild (8–14), moderate (15–23), and severe (24–64). This study used the highly reliable Korean version of HARS [30]. The K–HARS of this study achieved a high reliability (Cronbach’s α = 0.98).

The STAI was used to evaluate the anxiety level of the subjects [31]. It is a self-report questionnaire created to measure a person’s level of anxiety. The STAI consists of two forms. The STAI-S measures anxiety in the present moment (20 items, state anxiety), and the STAI-T measures anxiety levels as a personal characteristic (20 items, trait anxiety). In this study, we used STAI-T. This scale has 20 items, each with a four-point Likert scale (1–4). Higher scores indicate higher levels of anxiety. We employed the highly reliable Korean version of the STAI-T [32]. This study also showed high reliability (Cronbach’s α = 0.95). Since depression and anxiety disorder often occur together, it is vital to measure anxiety in patients with depression. Therefore, despite already measuring anxiety disorders with HARS, STAI was applied to the questionnaire. In addition, since HARS is for the clinician to evaluate the patient’s current anxiety symptoms, STAI-T was applied during STAI to evaluate the anxiety characteristics felt by the patients themselves.

### 2.7. Data Analysis

Data analysis was conducted using SPSS 18.0 Windows (SPSS, Chicago, IL, USA).

The paired t-test was used to compare the improvement in depression and anxiety before and after in each group (urban forest healing program and control). The group difference in benefits was compared by covariance analysis (ANCOVA). The Bonferroni test verified the post hoc analysis of group differences if significant differences were found in the covariance analysis. A *p*-value of less than 0.05 was considered statistically significant.

## 3. Results

### 3.1. Montgomery-Asberg Depression Rating Scale (MADRS)

Table 1 shows the pre-and post-test MADRS score for each group. The MADRS score of the urban forest healing program group was significantly lower after than before (from 31.64 ± 2.27 to 10.45 ± 1.76; *t* = 7.524, *p* < 0.001). Also, there were significant changes in the control group subjects’ MADRS scores (from 29.00 ± 2.91 to 21.36 ± 2.80; *t* = 4.063, *p* < 0.001).

In order to increase the verification power of the difference in MADRS scores according to the program’s application, a covariance analysis was conducted in which the pre-test scores were controlled as covariates (Table 2). The results showed the urban forest healing program group had a significantly lower MADRS score than the control group (F = 18.775, *p* < 0.001).

### 3.2. Hamilton Anxiety Rating Scale (HARS)

Table 3 shows the pre-and post-test HARS scores for each group. The HARS score of the urban forest healing program group was significantly lower after than before (from 26.05 ± 2.27 to 6.05 ± 0.87; *t* = 8.898, *p* < 0.001). Also, there were significant changes in the control group subjects’ HARS scores (from 26.08 ± 2.73 to 18.00 ± 2.43; *t* = 5.502, *p* < 0.001).

In order to increase the verification power of the difference in HARS scores according to the program’s application, a covariance analysis was conducted in which the pre-test scores were controlled as covariates (Table 4). As a result, the urban forest healing program group had a significantly lower HARS score than the control group (F = 35.916, *p* < 0.001).

### 3.3. State-Trait Anxiety Inventory Form Trait (STAI-T)

Table 5 shows the pre-and post-test STAI-T score for each group. The STAI-T score of the urban forest healing program group was significantly lower after the treatment than before (from 57.18 ± 1.83 to 50.09 ± 1.90; *t* = 4.382, *p* < 0.001). However, there were no significant changes in the control group subjects’ STAI-T scores (from 57.20 ± 2.04 to 55.60 ± 2.35; *t* = 1.389, *p* < 0.178).

In order to increase the verification power of the difference in STAI-T scores according to the program’s application, a covariance analysis was conducted in which the pre-test scores were controlled as covariates (Table 6). The results showed the urban forest healing program group had a significantly lower STAI-T score than the control group (F = 8.017, *p* = 0.007).

## 4. Discussion

This study investigated the benefits of an urban forest healing program on the depression and anxiety symptoms of depressive patients. This study demonstrated that combining general medication and urban forest healing programs significantly improves depression and anxiety symptoms in depressive patients more than treatment with medication alone.

Specifically, this study showed that the urban forest healing program decreased the MADRS score, which indicates the severity of depressive symptoms in patients with depression. The control group who received treatment as usual during the experimental period also showed a significant decrease in the MADRS score. However, the control group did not change to “moderate depression“ in the pre-test (29.00 ± 2.91 score) and the post-test (21.36 ± 2.80 score). On the other hand, the urban forest healing program group changed from “moderate depression“ in the pre-test (31.64 ± 2.27 score) to “mild depression“ in the post-test (10.45 ± 1.76 score). In addition, even when comparing the differences between groups, the urban forest healing program showed a statistically significantly lower MADRS score than the control group. This is consistent with previous studies that the forest healing program improves depressive symptoms in patients with major depressive disorder more than general treatment as usual [33,34]. For example, Woo et al. [33] investigated the effectiveness of the forest healing program for patients with major depressive disorders by dividing them into four groups: a forest healing program group, a hospital program group, a forest bathing group, and a control group that performs treatment as usual. As a result, it was reported that the forest healing program group had a significantly lower MADRS score than the control group that performed only treatment as usual.

Our results also showed that the urban forest healing program decreased the HARS score, which indicates the severity of anxiety symptoms in depression patients. Subjects in the control group also significantly reduced their HARS scores. The control group scores changed from “severe” in the pre-test (26.08 ± 2.73 score) to “moderate” in the post-test (18.00 ± 2.43 score). On the other hand, the subjects in the urban forest healing program group improved from “severe” in the pre-test (26.05 ± 2.27 score) to “normal“ in the post-test (6.05 ± 0.87 score). This can be interpreted as the urban forest healing program group showing a treatment reaction as the total score of HARS decreased by more than half. On the other hand, the control group improved the HARS score through outpatient treatment for six weeks but did not show a treatment reaction.

This result is consistent with previous results in which cognitive behavioral therapy improved subjects’ anxiety symptoms [8,34]. To our knowledge, a HARS measurement tool has never been used to investigate the effect of nature-based intervention on anxiety symptoms. In this study, it is meaningful to provide new evidence that urban forest healing programs have alleviated anxiety symptoms in patients with depression using HARS, which clinicians can evaluate.

The present study indicated that the urban forest healing program decreased the STAI-T scores of depression patients. Our results coincide well with previous studies of university students [35] and chronic stroke patients [19], which showed that forest healing programs improved subjects’ anxiety.

It is thought that reaching the complete recovery of social and professional functions is difficult only with pharmacological treatment. Despite pharmacological development, the depression response rate to antidepressant administration was found to be 50–70%, and the remission rate was only 30% [36]. In addition, it is known that patients tend to be afraid of returning to their daily lives and rarely escape from dysfunction. It is already a habit to stay at home helplessly, even though depressive symptoms have improved. Studies show that combining non-pharmacological and pharmacological treatments is beneficial in solving low compliance with antidepressant therapy, one of the biggest obstacles to depression treatment [37,38]. Therefore, an optimized treatment method that combines pharmacological and non-pharmacological treatment is required for better results.

Our findings suggest the use of forest healing as a non-pharmacological treatment. Most importantly, patients can interact with various environmental factors, including fresh air and open space, and five sensory stimuli, namely visual, sound, scent, and touch, with forests and trees. Previous studies have reported that five-sensory stimulation in a forest environment is effective in physical and mental relaxation [39,40,41,42]. Increasing patients’ physical activity through forest healing programs is also likely effective in restoring patients’ function. Physical activity is perceived to improve mental health, minimize the side effects of drugs, and reduce depression in treating mental disorders such as depression [43,44,45]. In addition, the study was conducted in urban forests with excellent accessibility. Previous studies have reported that urban green spaces are a cost-effective, simple, and accessible way to prevent depression [46,47]. Accordingly, it is crucial to actively induce patients to use urban forests daily for health recovery.

Moreover, the forest environment can facilitate physical activity and psychological interaction. It is known that people can reconsider interpersonal problems in forest environments. The forest environment has resolved their sense of isolation by allowing them to connect through objects in nature and sharing their presence with empathy and support among the subjects. According to Kim et al. [48], the forest healing program promotes social interactions, such as forming intimacy between subjects and improving interpersonal relationships. Hendee and Brown [49] also stated that collective forest experiences can provide social interaction and enhance bonding. This means that the more individuals share their feelings with others, the more cohesive they become among the groups. Therefore, physical and psychological activities increase by inducing urban forest healing programs to interact with the forest environment, which has a significant therapeutic effect in this study.

However, this present study has limitations. First, the types and dosages of antidepressants the patients were taking in this study could not be identified. Future studies need to analyze the effects using dosages as covariates. Second, we were not able to identify specifically the usual treatments of subjects in the experimental and control groups during the six-week study period. In future studies, it will be necessary to closely assess the treatment schedule of each subject during the study period and discuss the results. Third, this study did not identify the subjects’ usual exposure to nature. In future studies, the frequency and time of natural exposure of the subjects will need to be investigated and analyzed in detail. Fourth, the present study showed a difference in gender ratio between the experimental and the control groups in the randomization process. There was no stratification of the randomization procedure, but, in future studies, it will be necessary to stratify gender to allocate the gender ratio equally. Fifth, this study did not investigate the long-term effects of urban forest healing programs. Future studies will require a follow-up of continuous effects over six months. Sixth, the experimental intervention effect of this study may have included many of the benefits of social contact between participants through various activities in the forest. Therefore, in future studies, it will be necessary to establish a control group that performs similar socio-emotional activities outside the forest. Seventh, we used G*Power to obtain 52 subjects for sample size calculation, but some subjects were eliminated during protocol implementation, resulting in 50 subjects finally enrolled, a number less than the intended 52. This may affect the interpretation of the results. In future studies, it will be necessary to recruit more subjects in consideration of the dropout rate.

Although these limitations exist, our findings provide evidence that urban forest programs can be used as non-pharmacological interventions to improve depression and anxiety symptoms in depressive patients, along with pharmacological treatments.

## 5. Conclusions

This study demonstrated that supporting forest healing programs using urban forests in addition to general treatment significantly lowers depression and anxiety symptoms more than providing only general treatment to patients with depression. These results show that urban forest healing programs can be used as non-pharmacological interventions combined with medication. Therefore, it is expected that urban forest healing programs will be actively utilized as regular and continuous treatments to relieve depression and anxiety symptoms in depressive patients. To do this, it will be necessary to develop a medical plan that allows depressive patients to interact with nature in urban green spaces.

## Figures and Tables

**Figure 1 healthcare-11-02766-f001:**
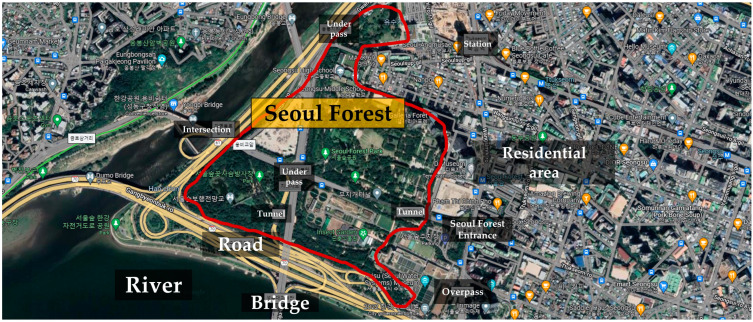
Study site: Seoul Forest (https://www.google.co.kr/maps/@/data=!3m1!1e3?hl=en) (accessed on 14 June 2023).

**Figure 2 healthcare-11-02766-f002:**
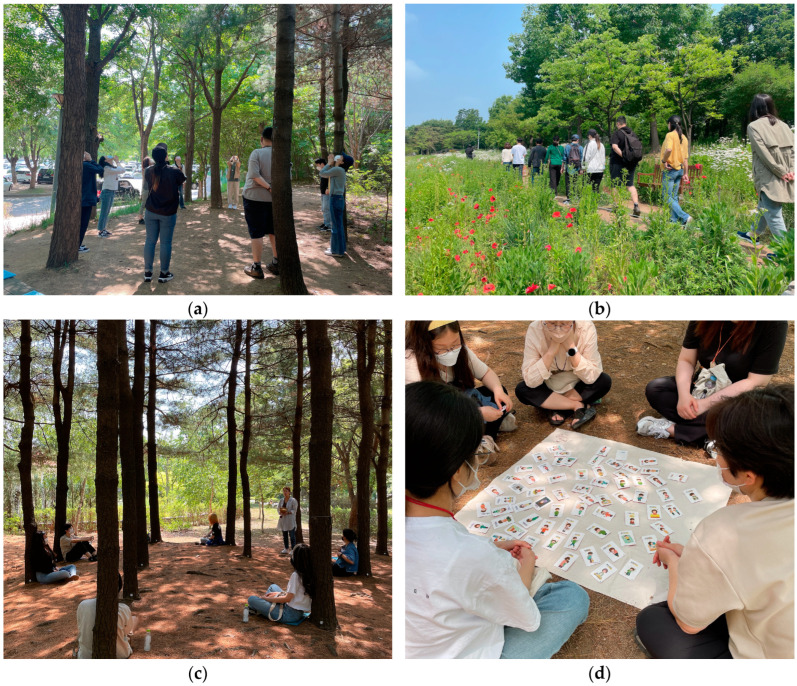
Activities of urban forest healing program. (**a**) stretching; (**b**) walking and the five senses (sound); (**c**) breathing; (**d**) playing an emotional card game.

**Table 1 healthcare-11-02766-t001:** Paired *t*-test analysis of Montgomery-Asberg depression rating scale.

Variable	UFHP (*n* = 22)				Cont. (*n* = 25)			
	Pre-Test	Post-Test	*t*	*p*	Pre-Test	Post-Test	*t*	*p*
	M (SE)	M (SE)			M (SE)	M (SE)		
MADRS	31.64 (2.27)	10.45 (1.76)	7.524	<0.001	29.00 (2.91)	21.36 (2.80)	4.063	<0.001

Note: UFHP, urban forest healing program group; Cont., control group.

**Table 2 healthcare-11-02766-t002:** Analysis of covariance of Montgomery-Asberg depression rating scale.

Variable	Sum of Squares	df	Mean Square	F	*p*
MADRS					
Pre-test	2045.669	1	2045.669	21.977	<0.001
Group	1747.599	1	1747.599	18.775	<0.001
Error	4095.545	44	93.081		

**Table 3 healthcare-11-02766-t003:** Paired *t*-test analysis of Hamilton Anxiety Rating Scale.

Variable	UFHP (*n* = 22)				Cont. (*n* = 25)			
	Pre-Test	Post-Test	*t*	*p*	Pre-Test	Post-Test	*t*	*p*
	M (SE)	M (SE)			M (SE)	M (SE)		
HARS	26.05 (2.27)	6.05 (0.87)	8.898	<0.001	26.08 (2.73)	18.00 (2.43)	5.502	<0.001

Note: UFHP, urban forest healing program group; Cont., control group.

**Table 4 healthcare-11-02766-t004:** Analysis of covariance of Hamilton Anxiety Rating Scale.

Variable	Sum of Squares	df	Mean Square	F	*p*
HARS					
Pre-test	1850.317	1	1850.317	39.857	<0.001
Group	1667.346	1	1667.346	35.916	<0.001
Error	2042.637	44	46.424		

**Table 5 healthcare-11-02766-t005:** Paired *t*-test analysis of State-Trait Anxiety Inventory form trait.

Variable	UFHP (*n* = 22)				Cont. (*n* = 25)			
	Pre-Test	Post-Test	*t*	*p*	Pre-Test	Post-Test	*t*	*p*
	M (SE)	M (SE)			M (SE)	M (SE)		
STAI-T	57.18 (1.83)	50.09 (1.90)	4.382	<0.001	57.20 (2.04)	55.60 (2.35)	1.389	0.178

Note: UFHP, urban forest healing program group; Cont., control group

**Table 6 healthcare-11-02766-t006:** Analysis of covariance of State-Trait Anxiety Inventory form trait.

Variable	Sum of Squares	df	Mean Square	F	*p*
STAI-T					
Pre-test	3053.757	1	3053.757	69.330	<0.001
Group	353.123	1	353.123	8.017	0.007
Error	1938.061	44	44.047		

## Data Availability

Not applicable.

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
