# Peer review of "Benefits of Urban Forest Healing Program on Depression and Anxiety Symptoms in Depressive Patients"

_healthcare, 2023, doi:10.3390/healthcare11202766_

Round 1
Reviewer 1 Report
Line 37. The word anhedonia should perhaps be in lower case?
Line 85-86. It should not be a weakness of previous studies that depressive symptoms are not assessed by psychiatrists. There are reliable and easy-to-use tests that do not require professional clinical supervision. If the authors understand that their work assesses mental health symptoms more reliably, they should refer to how they assess them, with what instruments and tools, what substantially differentiates their assessment mode from previous studies, and not only by what kind of professionals.
Line 88. Revise the wording. It is unclear whether ("it") refers to previous studies or to the current work.
In general this whole paragraph should be improved, especially at the end, where the objectives of the study should be described in more detail.
This last paragraph of the introduction should also indicate in what sense and to what extent their assessments (described in section 2.2 subjects) improve on those of the studies cited in the introduction.
Line 120. Could you elaborate on exclusion criterion (b)? What does double desease mean? do you mean allergy?
Line 148-150. Please say how many therapists were involved in total. Please also explain how many consultants were involved and in what role.
Line 131. They should explain what was done with the outpatients in the control group. Authors say that the purpose and procedure was explained to them; were they then assigned to a waiting list? On what pretext were they informed of a programme in which they were not going to participate?
Lines 142/143. Please do not repeat yourselves. You have just said the same thing four lines earlier
I don't think I have seen a description of who, when and where the pre-post measures were taken.
In order to properly assess the results, it would be very useful to know how many of the subjects, both in the experimental and control groups, were taking medication and how many received psychotherapy during the six-week programme. I think I understood that, but perhaps it should be stated more specifically whether the urban forest programme group also continued with its usual treatment and what this consisted of. It is important to know how many people in both groups were taking antidepressant medication, given that they suffered from moderate and mild depression, and perhaps not all were on medication. Does "All subjects" in line 165 refer to both groups?
I do not understand why meditation is mentioned in line 260, at the beginning of the discussion, as this concept does not appear in the introduction to the article, except as one of the components of the forest program.
Line 288. How do the authors measure whether or not there has been a treatment reaction?
While acknowledging the possible effect of social contact and group activity, the authors must recognise that they have not differentiated the possible socialisation benefits intrinsic to the programme. A control group that engaged in similar social-emotional activities outside the forest should have been used.
Included above
Author Response
Dear Editor and reviewers,
We would like to express our sincere gratitude for your kind consideration and comments on our manuscript. According to reviewers’ comments and suggestions, we revised the manuscript as follows:
(We marked the revision to the reviewer's comment in red)
- We unified ‘anhedonia’ into lower case (line 37).
- We believe that the previous study did not reveal the direct link between forest healing and depression because it investigated the effect of forest healing on depression in disease groups or health groups other than depression patients. So, we added the practical difference between our research and existing research (line 86-93).
- We added research objectives and hypothesis settings (line 86-100).
- We revised the exclusion criteria (b) (line 129-130).
- We added an explanation of how many therapists and counselors participated in this study and what role they played (line 171-182)
- We received consent by announcing that the forest healing program will be conducted in the first and second stages so that participants do not know whether it is an experimental or control group. In addition, the same forest healing program was provided to the control group after the end of the research period. We added this description (line 140-142, 185-187).
- We added a description of the pre- and post-evaluation (190-198).
- We could not identify the type and dose of antidepressants that many of the subjects in the experimental and control groups took during the six-week program. In addition, it was impossible to grasp the usual treatment during the study period specifically. This is the limitation of our research. So, we mentioned the need for further research and the limitations (line 366-372).
- We mislabeled medication as meditation. We modified it (line 292-293).
- The psychiatrist determined if there was a treatment response. Therefore, it was judged that there was a treatment response when the scale's total score was reduced by more than half through treatment.
- The experimental intervention effect of this study may have included much of the benefits of social contact between participants through various activities in the forest. We added to this point (line 379-383).
Reviewer 2 Report
this is an interesting study evalueting the efficacy of a non-medical treatment on depression levels. I think the manuscript can benefint from some revisions before it can be considered for publicarion. I report my comments/suggestions.
Introduction: the section is short and this may due to the limited study on the specific thematic. However, the authors may considerer to include some studies exploring other non-medical approach to depression treatment (see Verrusio et al 2014 DOI: 10.1016/j.ctim.2014.05.012 and Verrusio et al 2018
DOI10.1007/s12603-018-1044-2)Moreover, I suggest to add a paragraph exposing aims and hypothesis
METHODS-Subjects: the authors correctly reported the use of g power for sample size calculation, however during the protocol implementation some subjects drop out and the final number of participants was lower than required. I think this poin should be better discussed and considered through the manuscript.Data Analysis: I think the authors should also evaluate the baseline homogeneity between groups. Specifically they should evaluate if the groups were not statistically different for important dimensions as age, gender distribution, and psychological dimensions explored. This analysis is not clearly reported in the manuscript and should be added considering its relevance in the interpretation of results (only a difference in gender distribution was reported in the limitations).
Conclusion: please in this section more deeply explore the clinical relevance of present findings.
Minor editing
Author Response
Dear Editor and reviewers,
We would like to express our sincere gratitude for your kind consideration and comments on our manuscript. According to reviewers’ comments and suggestions, we revised the manuscript as follows:
(We marked the revision to the reviewer's comment in blue)
- We added a case for non-pharmacological intervention (line 54-60).
- We added research objectives and hypothesis settings (line 86-100).
- We added to the limitations that the number of subjects enrolled was less than the number required by the sample size calculation (line 383-386).
- We could not collect data on socioeconomic demographic characteristics other than age and gender for participants. This is a major limitation of our research. We will take note of your comments and supplement them in future studies.
- We added to the conclusion about the clinical relevance of the present results (line 396-400).
Reviewer 3 Report
This paper provides an additional confirmation of the value of forest healing programs on mitigating the symptoms of anxiety and depression. In both the Introduction and Discussion, the authors cite other studies that have made this link, but point out that those studies primarily included subjects who suffered from separate primary conditions, such as cancer, with anxiety and/or depression manifesting as secondary conditions. This justifies the need to report on the present study.
My biggest methodological concern, which the authors cite in the Limitations section, is the gender ratio imbalance between the urban forest healing group (7 M, 15 F) and the control group (1 M, 24 F). This gender disparity could have significantly affected the behavior of the test subjects and the authors should have taken measures to ensure something closer to gender balance between the groups.
Given the number of other limitations the authors cite, I recommend that this be titled “A Preliminary Stud
As with many papers I review written by researchers for whom English is not a first language, there are numerous examples of awkward sentence structure or confusing phrasing. The text would be substantially improved by being edited by an English speaker prior to publication.
I will reiterate that a thorough editing by a native English speaker is needed before this paper can be published.
Author Response
Dear Editor and reviewers,
We would like to express our sincere gratitude for your kind consideration and comments on our manuscript. According to reviewers’ comments and suggestions, we revised the manuscript as follows:
(We marked the revision to the reviewer's comment in Green)
- Our study showed significant gender ratio differences during randomization. We think this is a major limitation of our research. In retrospect, it would have been useful to stratify gender to equalize the number of men and women. In future research, we will stratify gender based on your comments (line 374-377).
- We received English proofreading thanks to your comment.
Round 2
Reviewer 1 Report
Please replace the word alcoholics (line 90) with something less stigmatising, e.g. alcohol abuse.
Be careful with the spelling. For example, line 155, conposition (sic) and unban (sic).
Check the language throughout the article. For example, depression questionnaire is sometimes called MADRAS and sometimes MADRS.
Please replace the word alcoholics (line 90) with something less stigmatising, e.g. alcohol abuse.
Be careful with the spelling. For example, line 155, conposition (sic) and unban (sic).
Check the language throughout the article. For example, depression questionnaire is sometimes called MADRAS and sometimes MADRS.
Author Response
Dear Editor and reviewers,
We would like to express our sincere gratitude for your kind consideration and comments on our manuscript. According to reviewers’ comments and suggestions, we revised the manuscript as follows:
(We marked the revision to the reviewer's comment in red)
- We modified ‘alcoholics’ to ‘alcohol abuse’ (Line 75, 90).
- We checked and corrected the spelling of the words.